# Perceived Social Support from Parents, Teachers, and Friends as Predictors of Test Anxiety in Chinese Final-Year High School Students: The Mediating Role of Academic Buoyancy

**DOI:** 10.3390/bs15040449

**Published:** 2025-04-01

**Authors:** Danwei Li, Nor Aniza Ahmad, Samsilah Roslan

**Affiliations:** Faculty of Educational Studies, Universiti Putra Malaysia, Serdang 43400, Malaysiasamsilah@upm.edu.my (S.R.)

**Keywords:** perceived social support, emotional support, academic buoyancy, test anxiety, Chinese final-year high school students

## Abstract

A pervasive and significant academic challenge confronted by students on a global scale is the phenomenon of test anxiety. This phenomenon is exacerbated in China, especially among final-year high school students who face college entrance exams. Perceived social support is widely regarded as the most prevalent protective factor against test anxiety. Academic buoyancy also demonstrates a significant correlation with test anxiety. However, there has been limited research on the potential relationship between perceived social support, academic buoyancy, and test anxiety. The purpose of this study is to examine the effects of specific sources and types of students perceived social support (e.g., emotional support from parents, teachers, and friends) on test anxiety and examine whether academic buoyancy serves as a mediating variable in the relationship between perceived social support and test anxiety. A total of 565 final-year high school students were selected as respondents from Heilongjiang Province in China. The result of SEM analysis indicated that the three sources of student-perceived emotional support could not predict test anxiety directly, but the students perceived three sources emotional support may have an indirect effect on test anxiety through the mediating role of academic buoyancy. In particular, perceived friend emotional support is the most beneficial among these sources of support for students. The anticipated outcomes of this study are expected to provide educators, counselors, and parents with key insights into the factors that alleviate test anxiety in high school students.

## 1. Introduction

Test anxiety has emerged as one of the most significant challenges faced by students globally ([21]). There is a growing consensus among researchers that the prevalence of test anxiety is rising ([9]). Test anxiety is a particularly salient issue in China. Recent studies have revealed that around 30 percent of Chinese high school students exhibit significant test anxiety, a trend that has remained consistent in recent years ([17]). This phenomenon is most evident among final-year high school students, as college entrance exams have a significant impact on their future career opportunities. Test anxiety manifests through various cognitive symptoms, including comparing one’s performance to peers, concern about the ramifications of test failure, and diminished self-efficacy in performance abilities ([19]). The impacts of test anxiety are extensive and can significantly influence the academic achievement of students and their well-being ([11]). Test anxiety may result in poor academic performance, decreased grade point averages, and increased dropout rates among affected student groups ([12]). Additionally, in severe cases, test anxiety may contribute to the development of serious psychological disorders and even suicidal behavior ([8]). Given these considerable impacts of test anxiety, it is important to investigate its factors, especially for the final-year high school students in China.

Perceived social support is often described as an important factor that protects people from high stress levels. According to social cognition theory ([2]), individuals’ thoughts, beliefs, and social environments interact to shape their behaviors and emotional responses. In the context of stress, perceived social support can influence how individuals perceive and appraise stressful events, thereby moderating the intensity of their stress reactions. The presence of supportive relationships, such as encouragement from friends, guidance from teachers, or reassurance from family members, can alleviate the pressure these individuals feel ([41]). Several studies have found a correlation between students’ perceived support from parents, teachers, and friends with their test anxiety ([1]; [20]; [24]; [36]; [42]); they identified social support as a mitigating factor for students’ test anxiety. However, the existing comprehension of the precise function of perceived social support in connection to test anxiety is still limited.

Parents, teachers, and friends are the three most influential social support sources in teenagers’ development and learning ([39]). It is widely recognized that parents, teachers, and peers play a crucial role during adolescence; however, only a limited number of studies have empirically compared the relative contributions of these three sources of perceived social support. Most research indicates that parental and teacher support play a predominant role in the academic development and well-being of students, while the influence of peer support is considered to be relatively minor ([7]; [32]; [38]; [40]). However, as students’ progress in age, the factors influencing friend support become increasingly significant and cannot be overlooked. This can be attributed to adolescents transcending their family sphere and entering broader social contexts ([30]). Therefore, for students in their final year of high school, the individual contributions of these three sources of support warrant further investigation.

Perceived emotional support refers to an individual’s belief that he or she will receive emotional encouragement when needed ([6]), which is a type of support that is closely associated with students’ academic development ([31]). To date, there is less research on the relationship between different sources of emotional support and test anxiety. For the perceived parental emotional support, [25] ([25]) examined the relationship between perceived parental emotional support with students’ test anxiety, and they found that perceived parental emotional support was identified as a protective factor that positively influenced students’ mental health. Similarly, [32] ([32]) drew the same conclusion, they revealed that perceived parental emotional support emerged as the most beneficial factor, predicting a reduction in test anxiety. For the perceived teacher emotional support, [29] ([29]) demonstrated that the perceived emotional support from teachers was negatively associated with academic anxiety. Similarly, [18] ([18]) revealed that students who perceived more relational support from their teachers had lower levels of test anxiety. For the perceived friend emotional support, limited research has been conducted between friend emotional support and test anxiety. Additional investigation is required to elucidate the impact of perceived friend emotional support on test anxiety.

However, while social support is generally seen as beneficial, the relationship between perceived emotional support from different sources and test anxiety remains a subject of debate in the literature. For example, well-meaning encouragement from parents or teachers may be perceived as pressure to succeed, leading to increased anxiety rather than alleviating it ([1]; [27]). Additionally, friend support might create a sense of competition or comparison, which could intensify feelings of inadequacy and stress; research has demonstrated that higher levels of friend support are associated with elevated negative emotional states among students ([30]). Therefore, more investigation is required to clarify the underlying mechanisms between perceived emotional support from different sources and test anxiety.

Academic buoyancy represents a relatively novel notion within the field of positive psychology. A student’s academic buoyancy is determined by their ability to conquer the academic obstacles in school life, such as managing workloads, handling deadlines, and dealing with occasional setbacks ([26]). According to social cognition theory ([2]), good perceived social support helps to enhance students’ ability (i.e., academic buoyancy) to cope with their academic challenges. In other words, when students perceive support from others, they are more inclined to develop confidence in their ability to handle academic demands, which in turn reduces their susceptibility to stress and anxiety, particularly in testing situations ([26]). Some studies have demonstrated that academic buoyancy could function as a mediating variable between perceived social support and other variables ([3]; [13]; [16]; [22]). For example, [43] ([43]) reported that academic buoyancy was a mediating variable between teacher–student relationship support with academic achievement. In addition, according to [28] ([28]), academic buoyancy was found to be a mediating variable between parental support with academic engagement. Meanwhile, some studies also used academic buoyancy as a mediating variable that looked at test anxiety as an outcome. For example, [35] ([35]) conducted a study in the US and Turkey, and they reported that there was a negative correlation between academic buoyancy and test anxiety. In addition, [24] ([24]) proved that the mediation of academic buoyancy assisted in predicting high school students’ test anxiety. By integrating these two areas of research into a single study, it becomes feasible to propose that academic buoyancy serves as a mediating factor in the relationship between perceived social support and test anxiety.

Considering the aforementioned context, this study aims to determine the relationship between perceived social support (e.g., perceived emotional support from parents, teachers, and friends), academic buoyancy, and test anxiety, and whether academic buoyancy can play the mediating role between perceived social support and test anxiety. To our knowledge, no studies have been undertaken on the potential impact of perceived social support on academic buoyancy and test anxiety. The anticipated outcomes of this research are expected to inform educators, counselors, and policymakers about the critical factors that help mitigate test anxiety in high school students. The current study offers the following hypotheses (see Figure 1).

**H1.** 
*There is a significant association between perceived parental emotional support and test anxiety.*


**H2.** 
*There is a significant association between perceived teacher emotional support and test anxiety.*


**H3.** 
*There is a significant association between perceived friend emotional support and test anxiety.*


**H4.** 
*There is a significant association between perceived parental emotional support and academic buoyancy.*


**H5.** 
*There is a significant association between perceived teacher emotional support and academic buoyancy.*


**H6.** 
*There is a significant association between perceived friend emotional support and academic buoyancy.*


**H7.** 
*There is a significant association between academic buoyancy and test anxiety.*


**H8.** 
*Academic buoyancy serves as a mediator between perceived parental emotional support and test anxiety.*


**H9.** 
*Academic buoyancy serves as a mediator between perceived teacher emotional support and test anxiety.*


**H10.** 
*Academic buoyancy serves as a mediator between perceived friend emotional support and test anxiety.*


## 2. Methodology

### 2.1. Participants and Procedures

This study’s targeted population comprises all public final-year high school students in Heilongjiang Province, China. The researcher used the proportional stratified sampling strategy. First, the researcher stratified according to grade level, only selecting final-year high school students in Heilongjiang Province. The next step was to calculate the number of students required from each city proportionately. In 2023, there were 189,235 final-year high school students in 13 cities in Heilongjiang Province. The required minimum number of students for each city is calculated based on the proportion of final-year high school students in the 13 cities of Heilongjiang Province. Due to 10 randomly selected samples can adequately reflect the attitudes and perspectives of the group ([5]), the researchers recruited more than 10 students from each school to ensure the effectiveness of questionnaire collection. The population for this research is 189,235, which is more than 100,000, and the minimum required sample size should be 384 ([23]); therefore, the number of schools to be surveyed should be around 40. To ensure the random process, all schools and students from the schools were selected through simple random sampling in each city. Ultimately, a total of 565 final-year high school students from 41 schools in Heilongjiang Province, China, participated in this study. Most of the respondents’ ages was 18 (97.2%). The proportion of male and female were about equal; 46.5% of the respondents were males, and 53.5% of the respondents were females. And the proportion of respondents from urban and rural areas was equally divided (45.5% and 54.5%).

### 2.2. Measures

The data collection process utilized a well-defined instrument structured into two distinct sections. Section A focused on obtaining demographic information from respondents, including gender, age, residence registration location. Section B was designed to collect data on perceived social support, academic buoyancy, and test anxiety.

#### 2.2.1. Perceived Emotional Support Scale

The Perceived Social Emotional Scale was adapted from three questionnaires (Perceived Parental Academic Support Scale, Perceived Teacher Academic Support Scale, and Perceived Friend/Peer Academic Support Scale) developed by [6] ([6]). Only the emotional support dimensions of each questionnaire were selected and adapted for this scale; therefore, this scale contains three dimensions: perceived parental emotional support; perceived teacher emotional support; and perceived friend emotional support. This scale has a total of 21 items, and it is a 5-point Likert scale (1 = strongly disagree, 5 = strongly agree). No items are reverse scored. A higher overall score indicates a higher level of students perceived social support. The current Cronbach Alpha reliability coefficient for this scale was 0.944, and the three dimensions were 0.929, 0.922 and 0.915, respectively. The CFA model fitness was met at Chi-square = 274.639; df = 177; *p* = 0.000; χ^2^/df = 1.552; RMR = 0.032; NFI = 0.967; RFI = 0.961; GFI = 0.957; CFI = 0.988; IFI = 0.988; TLI = 0.986; AGFI = 0.943; RMSEA = 0.031.

#### 2.2.2. Academic Buoyancy Scale

The Chinese version of the Academic Buoyancy Scale compiled by [34] ([34]) was employed to measure academic buoyancy. The original version of the questionnaire was ABS compiled by [26] ([26]). The responses on the scale were calculated on a 5-point Likert scale (1 = strongly disagree, 5 = strongly agree). This scale only has one dimension, and no items are reverse scored. A higher overall score indicates a higher level of academic buoyancy. The current Cronbach Alpha reliability coefficient for this scale was 0.916. The CFA model fitness was met at Chi-square = 6.251; df = 2; *p* = 0.044; χ^2^/df = 3.126; RMR = 0.012; NFI = 0.996; RFI = 0.988; GFI = 0.995; CFI = 0.997; IFI = 0.997; TLI = 0.992; AGFI = 0.973; RMSEA = 0.061.

#### 2.2.3. Brief FRIEDBEN Test Anxiety Scale

Test anxiety was be assessed using the Brief- FRIEDBEN Test Anxiety Scale compiled by [37] ([37]). The original version of the questionnaire included 12 items; after expert review, this questionnaire was revised into 13 items. This scale is a 5-point Likert scale (1 = strongly disagree, 5 = strongly agree). A higher overall score indicates a higher level of test anxiety. This scale has three subscales that focus on crucial behavioral symptomatologic expressions: tenseness, social derogation, and cognitive obstruction. The current Cronbach Alpha reliability coefficient for this scale was 0.904. The CFA model fitness was met at Chi-square = 133.736; df = 58; *p* = 0.000; χ^2^/df = 2.306; RMR = 0.056; NFI = 0.964; RFI = 0.951; GFI = 0.963; CFI = 0.979; IFI = 0.979; TLI = 0.972; AGFI = 0.942; RMSEA = 0.048.

### 2.3. Data Analysis

SPSS version 26 was used to analyze the descriptive statistical data, while AMOS version 26 was employed to conduct hypothesis testing analyses. The evaluation of model fit was conducted using the following goodness-of-fit indices and their respective thresholds: χ^2^/df ≤ 5, GFI ≥ 0.90, CFI ≥ 0.90, NFI ≥ 0.90, TLI ≥ 0.90, RMSEA ≤ 0.08, and SRMR ≤ 0.10.

## 3. Results

### 3.1. Descriptive Analysis and Correlation Analysis

As shown in Table 1, the findings from the descriptive analysis revealed that students perceived a high level of social emotional support from their parents, teachers, and friends (3.95, 3.93, and 3.67, respectively) and had a moderate level of academic buoyancy (3.44) and test anxiety (2.75). The interpretations of the mean scores are based on Ho Robert ([15]). Table 1 also displays the intercorrelations between the factors evaluated for Chinese final-year high school students. Perceived parental emotional support (PPES) was significantly and positively associated with perceived teacher emotional support (PTES) (r = 0.559, *p* < 0.01) and perceived friend emotional support (PFES) (r = 0.535, *p* < 0.01). PTES correlated significantly and positively with PFES (r = 0.524, *p* < 0.01). There was a significant positive relationship between PPES and academic buoyancy (r = 0.379, *p* < 0.01), PTES and academic buoyancy (r = 0.359, *p* < 0.01), and PFES and academic buoyancy (r = 0.383, *p* < 0.01). Test anxiety was significantly and negatively associated with PPES (r = −0.246, *p* < 0.01), PTES (r = −0.204, *p* < 0.01), PFES (r = −0.262, *p* < 0.01), and academic buoyancy (r = −0.431, *p* < 0.01).

### 3.2. Results of Structural Equation Modeling

This study employed SEM to test hypotheses. The assessment of the structural model fit yielded a chi-square value of 1403.711 with 645 *df*, and a *p*-value of 000, indicating a significant model fit. The goodness-of-fit indices were as follows: GFI = 0.881, AGFI = 0.863, CFI = 0.945, NFI = 0.903, RFI = 0.894, IFI = 0.945, and TLI = 0.940. The RMSEA was reported at 0.046, and the relative CMIN/*df* was 2.176. Table 2 and Figure 2 display the results for the path coefficient. Findings demonstrated that PPES, PTES, and PFES were not found to significantly predict test anxiety, respectively. Thus, H1, H2, and H3 were rejected. In addition, PPES, PTES, and PFES were found to significantly and positively predict academic buoyancy, respectively (β = 0.224, *p* < 0.001; β = 0.173, *p* < 0.001; β = 0.287, *p* < 0.001, respectively). Thus, H4, H5, and H6 were accepted. Furthermore, academic buoyancy was a significant negative predictor of test anxiety (β = −0.359, *p* < 0.001). Therefore, H7 was accepted.

### 3.3. The Mediating Effect of Academic Buoyancy

Bootstrapping was used in AMOS to test the mediation effect, providing a re-confirmation of the mediation results at a 95%confidence interval to check the significance of the mediation effect and compare the magnitude of mediation outcomes. Table 3 shows the results of examining the mediating effect of academic buoyancy on the relationship between perceived social support and test anxiety. The bootstrap test revealed that the 95% confidence intervals for the three paths did not contain 0, showing that the total effect was significant (effect = −0.391, 95% CI: −0.511, −0.272). In addition, the total indirect effect was also significant (effect = −0.245, 95% CI: −0.340, −0.170). Perceived social support affected the test anxiety of Chinese final-year high school students through three intermediary paths: Indp1 = PPES → academic buoyancy → test anxiety (effect = −0.080, 95% CI: −0.151, −0.031); Indp2 = PTES → academic buoyancy → test anxiety (effect = −0.062, 95% CI: −0.116, −0.022); Indp3 = PFES → academic buoyancy → test anxiety (effect = −0.103, 95% CI: −0.173, −0.049). Therefore, H8, H9, and H10 were supported. Meanwhile, the magnitudes of the three mediating effects are ranked as follows: Indp3 > Indp1 > Indp2.

## 4. Discussion

The research findings indicated that perceived parental emotional support, perceived teacher emotional support, and perceived friend emotional support were not found to significantly predict test anxiety, respectively. This is inconsistent with previous research findings ([18]; [25]; [29]; [32]). Several factors may account for this result: First of all, perceived emotional support primarily provides care, active listening, and encouragement. While perceived emotional support can offer psychological comfort, it may not directly assist students in addressing test-related practical issues. In addition, considering mediating or moderating variables, the direct predictive effect of emotional support from different sources on test anxiety is relatively weak and may indirectly influence test anxiety through other pathways. Furthermore, the subjects in previous studies were relatively younger ([29]; [32]), for example, elementary and middle school students, whereas the participants in this study were final-year high school students. For the final-year high school students, they are in a stage of seeking independence, and excessive emotional support may be perceived as interference or control. As a result, in this study, perceived social support did not directly exhibit a significant impact on test anxiety.

This study showed that academic buoyancy was the mediator between perceived social support and test anxiety. This result is consistent with previous research findings ([20]; [24]). Students exhibiting higher levels of academic buoyancy are more likely to have strong social resources, and these accumulated resources can be drawn upon as needed, thus enhancing an individual’s resilience to stress and adversity. When encountering academic setbacks and challenges, this approach could enhance students’ experiences of academic enjoyment and hopefulness, while simultaneously reducing feelings of academic anxiety, boredom, and hopelessness ([14]).This finding indicates that perceived social support had a crucial role in reducing test anxiety by enhancing students’ academic buoyancy. According to social cognition theory ([2]), learning occurs within a social context, and students’ perceived social support can impact their cognition, behaviors, and emotions. Perceived social support appears to bolster students’ academic buoyancy, which serves as a psychological buffer mitigating the negative impacts of test anxiety. This buffering mechanism highlights the importance of both external resources (perceived social support) and internal capacities (academic buoyancy) in managing academic anxiety.

This study also found that the impact of perceived emotional support from friends had the strongest influence on test anxiety among the three sources of support, followed by perceived emotional support from parents, and lastly, perceived emotional support from teachers. This result aligns with previous research ([4]), which found that among students aged 16–18, support from friends exceeded that from parents, while support from teachers was lower for older age groups, likely due to the transition from primary to secondary school. The majority of participants in this study were 18 years old, and as individuals mature psychologically, the quality of adolescent friendships improves, and over time, friends become increasingly capable of providing emotional support to one another. Furthermore, this study highlights that perceived emotional support from parents remains highly significant for adolescents. This finding is consistent with the previous literature (e.g., [32]; [33]). [30] ([30]) also demonstrated the importance of parental support across different age groups, underscoring the central role of parent–adolescent relationships in adolescent development. The relatively lower contribution of perceived emotional support from teachers to students can be attributed to several factors. First, the heavy academic workload in high school may reduce the time available for teacher–student interactions. Additionally, exam-oriented teaching approaches may weaken opportunities for emotional exchange. Moreover, as final-year high school students transition into late adolescence, they develop greater autonomy and independence, leading to a natural decline in their reliance on adult authority figures, including teachers ([10]). These changes collectively influence the effectiveness of perceived emotional support from teachers.

There are some theoretical implications for the field of educational psychology from this work. Firstly, this study proposes a new and comprehensive framework that encompasses perceived social support, academic buoyancy, and test anxiety. This framework not only introduces new perspectives for academic research but also provides practical tools for educational practice, enabling educators and parents to better understand and support students in their learning processes. Secondly, this study expands the relevant body of knowledge. First of all, this research is expected to fill the gap by providing some valid knowledge on the existing research controversies in the relationship between perceived social support, academic buoyancy, and test anxiety. Moreover, this research enhances the existing literature on academic buoyancy by identifying it as a mediating variable between differentiated sources of social support (perceived emotional support from parents, teachers, and friends) with test anxiety.

This study also has several important implications for educational practice. Firstly, for educators and school administrators, this research emphasizes the importance of creating a supportive classroom environment. Emphasizing the provision of emotional support by teachers to students, as well as mutual emotional support among peers. By fostering an atmosphere that encourages open communication and reduces competitive pressures, educators can help mitigate students’ anxiety and enhance their academic performance. Secondly, for parents, this research highlights the importance of parental involvement in managing test anxiety. Parents can be educated on effective strategies for supporting their children, such as creating a warm and supportive home environment. By actively participating in their children’s academic lives and providing emotional support, parents can serve a pivotal role in alleviating test anxiety and enhancing overall student well-being. Lastly, one of the key significances of this study is to help students recognize the negative impacts of test anxiety and the importance of emotional support among friends. By revealing the specific negative consequences of test anxiety, such as reduced learning efficiency, increased psychological stress, and health problems, this research enables students to grasp the severity of these issues and motivates them to take action to manage and alleviate their anxiety. Meanwhile, friends’ interactions can provide understanding and empathy from a peer perspective, enabling the sharing of stressors and serving as a crucial channel for emotional expression and anxiety relief. This study identifies key factors influencing test anxiety, and these factors helps students understand which aspects might exacerbate their anxiety and provides them with targeted coping strategies.

Based on the results and these limitations, the following recommendations for future research are offered: First, future studies should prioritize conducting long-term longitudinal studies to better understand the dynamic relationships between perceived social emotional support, academic buoyancy, and test anxiety. Second, according to the results of this study, future research should conduct targeted intervention studies. These studies include the following: Design and test intervention programs aimed at enhancing perceived social support and increasing academic buoyancy. Evaluate the effectiveness of these interventions in reducing test anxiety. Third, given the variations in educational systems and cultural contexts, future research should replicate this study across diverse national and cultural settings and examine the moderating effects of cultural factors on the relationships among these study variables.

## 5. Conclusions

The objective of this research was to investigate the potential relationship between perceived social support, academic buoyancy, and test anxiety among Chinese final-year high school students. The results showed that academic buoyancy serves as a full mediator in the relationship between perceived social support and test anxiety. Specifically, the emotional support perceived by students from parents, teachers, and friends cannot directly predict their test anxiety; however, these supports can indirectly affect test anxiety through the mediating role of academic buoyancy. Notably, among the three sources of support, students perceive emotional support from friends as the most beneficial to them. By this study, policymakers, educators, parents, and students can obtain a deeper understanding of how to alleviate students’ test anxiety. More prospective studies are needed to confirm the dynamic relationships between perceived social support, academic buoyancy, and test anxiety.

## Figures and Tables

**Figure 1 behavsci-15-00449-f001:**
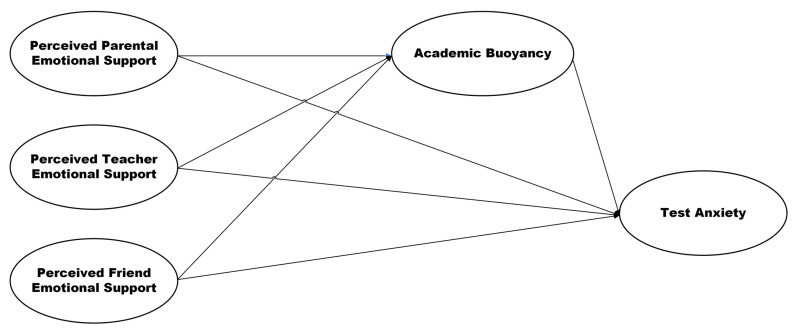
Hypothesized model of the associations between the study’s constructs.

**Figure 2 behavsci-15-00449-f002:**
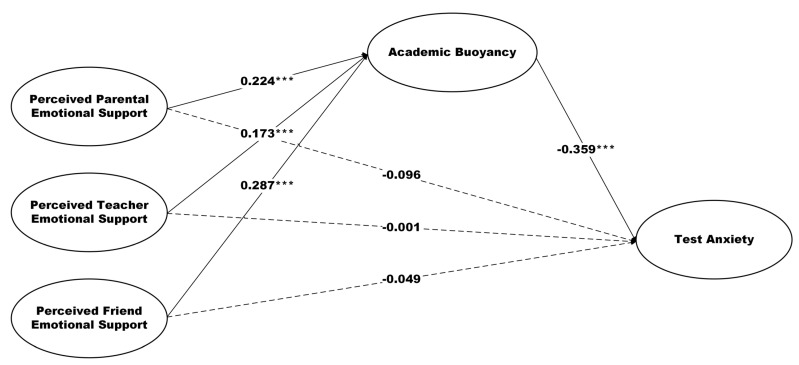
Path coefficient results. *** *p* < 0.001.

**Table 1 behavsci-15-00449-t001:** Descriptive analysis and correlation analysis (n = 565).

	M	SD	PPES	PTES	PFES	AB	TA
PPES	3.95	0.89	1				
PTES	3.93	0.84	0.559 **	1			
PFES	3.67	0.91	0.535 **	0.524 **	1		
AB	3.44	1.01	0.379 **	0.359 **	0.383 **	1	
TA	2.75	0.86	−0.246 **	−0.204 **	−0.262 **	−0.431 **	1

PPES = perceived parental emotional support; PTES = perceived teacher emotional support; PFES = perceived friend emotional support; AB = academic buoyancy; TA = test anxiety; ** *p* < 0.01.

**Table 2 behavsci-15-00449-t002:** The path coefficient of variables.

Hypothesized Relationship	Beta	S.E.	C.R.	*p*
H1: PPES → TA	−0.096	0.049	−1.959	0.050
H2: PTES → TA	−0.001	0.042	−0.014	0.989
H3: PFES → TA	−0.049	0.044	−1.119	0.263
H4: PPES → AB	0.224	0.047	4.801	***
H5: PTES → AB	0.173	0.044	3.924	***
H6: PFES → AB	0.287	0.052	5.564	***
H7: AB → TA	−0.359	0.050	−7.225	***

PPES = perceived parental emotional support; PTES = perceived teacher emotional support; PFES = perceived friend emotional support; AB = academic buoyancy; TA = test anxiety; Beta = standardized regression coefficient; S.E. = standard error; C.R. = critical ratio; *** *p* < 0.001.

**Table 3 behavsci-15-00449-t003:** Results of examining mediating effect of academic buoyancy.

Model/Hypothesized Path	Beta	S.E.	*p*	95% Confidence BC CI	
				LB	UB
Total effects	−0.391	0.061	***	−0.511	−0.272
Total indirect effects	−0.245	0.043	***	−0.340	−0.170
Indp1: PPES → AB → TA	−0.080	0.030	**	−0.151	−0.031
Indp2: PTES → AB → TA	−0.062	0.024	**	−0.116	−0.022
Indp3: PFES → AB → TA	−0.103	0.031	***	−0.173	−0.049

PPES = perceived parental emotional support; PTES = perceived teacher emotional support; PFES = perceived friend emotional support; AB = academic buoyancy; TA = test anxiety; Beta = standardized regression coefficient; S.E. = standard error; 95% CI BC: bias corrected at 95% confident interval; LB = lower boundary; UB = upper boundary; ** *p* < 0.05; *** *p* < 0.001.

## Data Availability

Data is contained within the article.

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
