# Peer review of "Perceived Social Support from Parents, Teachers, and Friends as Predictors of Test Anxiety in Chinese Final-Year High School Students: The Mediating Role of Academic Buoyancy"

_behavsci, 2025, doi:10.3390/bs15040449_

Round 1
Reviewer 1 Report
Comments and Suggestions for Authors
This is a strong paper, with clear hypothesise, supported by the literature and clear analysis. There are only a few suggestions for improvement as detailed below.
Methodology: I would like to know more about the following:
1. the stratified sampling - it is suggested that within the Province cities were selected - all cities or just some? Why not all schools within the Province sampled?
2. ll216-217 despite the sample being drawn from cities ll216-217 indicate that slightly more of the participants were from rural as opposed to urban backgrounds. Can you explain this if recruited from cities?
3. What time of year was data collected? Near exam times? I think it would be good to know as I expect this would impact on self-report measures such as these
4. Whilst the sample looks well balanced what proportion of the total population is 565 final year students? Any indications of how representative they are of the student population on other factors?
Discussion:
I was wondering if in the future research section you would think it important to have some more qualitative research understanding how these quantitatively measured mechanisms work in practice as your literature review indicates some uncertainty in how these forms of support actually operate e.g. ll56-57 there appears to be some uncertainty around how the support of friends actually works?
Some general tidying up of the English language throughout e.g. l121-22 'In particularly, the friends emotional support' change to 'In particular, friends emotional support', l43 'it is quite significant to investigate' change to 'it is important to investigate', l70 'to remain buoyancy' change to 'to remain buoyant' etc.
Please write abbreviations in full followed by brackets in the main text the first time they are used.
Comments on the Quality of English LanguagePlease see previous comments. Could do with improvement throughout - mainly surrounding tenses and choice of words. It's understandable but another sweep through would really help present your research more clearly.
Author Response
【Comments1-Methodology】
1. the stratified sampling - it is suggested that within the Province cities were selected - all cities or just some? Why not all schools within the Province sampled?
2. ll216-217 despite the sample being drawn from cities ll216-217 indicate that slightly more of the participants were from rural as opposed to urban backgrounds. Can you explain this if recruited from cities?
3. What time of year was data collected? Near exam times? I think it would be good to know as I expect this would impact on self-report measures such as these
4. Whilst the sample looks well balanced what proportion of the total population is 565 final year students? Any indications of how representative they are of the student population on other factors?】
Reply:
Dear reviewer, thank you for your constructive feedback.
For questions 1&4, I have provided a detailed addition in section 3.1 (Sample and Selection).
For question 2, no control was applied for whether students were from urban or rural areas, as all students were selected randomly from schools based on urban stratification. Consequently, the final results did not show significant differences. This issue arose due to a limited explanation regarding the sample selection process, which led to some misunderstandings.
For question 3, the survey was conducted in September 2023, coinciding with the beginning of the first semester of the final year of high school and is not close to the exam period. Since this study is cross-sectional, I did not specify a particular measurement time.
【Comments 2-Discussion】
I was wondering if in the future research section you would think it important to have some more qualitative research understanding how these quantitatively measured mechanisms work in practice as your literature review indicates some uncertainty in how these forms of support actually operate e.g. ll56-57 there appears to be some uncertainty around how the support of friends actually works?
Reply:
I believe that future research should include more qualitative studies, as the findings of this study suggest that social support has a significant indirect effect on test anxiety, particularly emotional support from friends, which has the greatest impact on students' test anxiety. The uncertainty surrounding the influence of "support from friends" on pages 56-57 reflects an existing controversy in the literature, and I believe it is appropriate to address this as a point of argument within this paper.
【Comments 3】
Some general tidying up of the English language throughout e.g. l121-22 'In particularly, the friends emotional support' change to 'In particular, friends emotional support', l43 'it is quite significant to investigate' change to 'it is important to investigate', l70 'to remain buoyancy' change to 'to remain buoyant' etc. Please write abbreviations in full followed by brackets in the main text the first time they are used.
Reply:
All of these issues have been addressed and highlighted in the revised manuscript. Thank you so much for your feedback.
Reviewer 2 Report
Comments and Suggestions for Authors
The manuscript describes the adaptation of various scales to measure the key constructs of the study. However, it does not mention whether a Confirmatory Factor Analysis (CFA) was conducted to validate the factorial structure of the adapted scales within the specific context of final-year high school students in China. This analysis would be crucial to ensure the validity of the scales, especially given that they were adapted from previous questionnaires. If conducting a CFA was not feasible, it would be important to clearly justify this decision in the manuscript and address it as a limitation of the study.
Author Response
Comments: The manuscript describes the adaptation of various scales to measure the key constructs of the study. However, it does not mention whether a Confirmatory Factor Analysis (CFA) was conducted to validate the factorial structure of the adapted scales within the specific context of final-year high school students in China. This analysis would be crucial to ensure the validity of the scales, especially given that they were adapted from previous questionnaires. If conducting a CFA was not feasible, it would be important to clearly justify this decision in the manuscript and address it as a limitation of the study.
Response: Dear reviewer, Hello! Thank you for your feedback. I have added all CFA data for all scales in Section 3.2 (Measurement). Thank you very much for your help.